# How to Manage Withdrawal of Sedation and Analgesia in Mechanically Ventilated COVID-19 Patients?

**DOI:** 10.3390/jcm10214917

**Published:** 2021-10-24

**Authors:** Amédée Ego, Katarina Halenarova, Jacques Creteur, Fabio Silvio Taccone

**Affiliations:** Department of Intensive Care, Erasme Hospital, Université Libre de Bruxelles (ULB), 1070 Brussels, Belgium; Katarina.Halenarova@erasme.ulb.ac.be (K.H.); Jacques.creteur@erasme.ulb.ac.be (J.C.); Fabio.taccone@erasme.ulb.ac.be (F.S.T.)

**Keywords:** sedation, analgesia, COVID-19, withdrawal, management

## Abstract

COVID-19 patients suffering from severe acute respiratory distress syndrome (ARDS) require mechanical ventilation (MV) for respiratory failure. To achieve these ventilatory goals, it has been observed that COVID-19 patients in particular require high regimens and prolonged use of sedatives, analgesics and neuromuscular blocking agents (NMBA). Withdrawal from analgo-sedation may induce a “drug withdrawal syndrome” (DWS), i.e., clinical symptoms of anxiety, tremor, agitation, hallucinations and vomiting, as a result of adrenergic activation and hyperalgesia. We describe the epidemiology, mechanisms leading to this syndrome and our strategies to prevent and treat it.

## 1. Introduction

COVID-19 patients suffering from severe acute respiratory distress syndrome (ARDS) who were admitted to the intensive care unit (ICU) often require mechanical ventilation (MV) for severe impairment of gas exchanges and respiratory failure [1]. In this setting, decreasing the work of breathing, applying lung-protective ventilation and limiting asynchronies minimize the risk of ventilator-induced lung injury (VILI) [2]. To achieve these ventilatory goals, it has been observed that COVID-19 patients require high regimens of sedatives, analgesics and neuromuscular blocking agents (NMBA) [3,4,5]. More importantly, due to a slow disease evolution, sedation is frequently prolonged for more than 7 days. Additionally, in these patients, there is a higher concern for an unintentional self-extubation, exposing them to the risk of severe hypoxemia until emergency intervention requiring personal protective equipment is performed; also, efforts are made to reduce the need for medical and nurse entry into the patient’s room, which contributes to the high sedation maintenance [3]. Finally, morbid obesity and organ dysfunction, frequently seen in COVID-19 patients, result in drug accumulation.

Deep sedation in patients with COVID-19, as in any severe ARDS, is associated with several complications, such as ventilation-associated pneumonia (VAP), gastroparesis (i.e., reduced enteral nutrition daily regimens) or intestinal hypomotility (i.e., abdominal distension and increased airway pressure), delirium and prolonged duration of MV and ICU-acquired weakness (ICU-AW) [6]. These complications further lead to prolonged ICU and hospital stay and, potentially, increased mortality.

In patients whose pulmonary function is restored over time, abrupt discontinuation or rapid reduction of sedatives and analgesics may induce a “drug withdrawal syndrome” (DWS), i.e., clinical symptoms of anxiety, tremor, agitation, hallucinations and vomiting, as a result of adrenergic activation and hyperalgesia. DWS frequently prompts physicians to readminister sedatives or analgesics and leads to even longer duration of MV [7].

## 2. Epidemiology

DWS is widely described in the neonatal and pediatric intensive care literature, and its incidence is particularly high, ranging from 42% to 95% of children receiving continuous infusion opioids and benzodiazepines [8,9,10]. A retrospective study conducted in adult patients showed that one third of patients undergoing sedation during the ICU stay experienced DWS [11]. In other small cohorts [12,13,14], the incidence of DWS varied between 17 and 100%.

In non-COVID-19 ICU patients, DWS has often been underestimated or misdiagnosed, due to generally lighter sedation administered for shorter periods of time. In this setting, the main identified risk factors for DWS are past medical history of prolonged ICU stay with sedatives or analgesics administration, prior substance dependence or chronic opioid/benzodiazepine use, but it may also occur in “drug-naïve” patients treated acutely with prolonged infusions of analgo-sedation, which makes the duration of the ICU stay a risk factor per se [12]. This syndrome has gained more attention with the COVID-19 pandemic because of prolonged sedation use, although no epidemiological data have been published so far on this topic.

## 3. Pathophysiology, Diagnosis and Clinical Presentation

The pathophysiology of DWS is related to the development of a “dependence” of the patients on the administered drug; as such, most of the related symptoms will depend on the specific receptors on which the drug binds, the total dose and the route of administration [7]. The abrupt interruption of one drug results in an imbalance of the equilibrium in the human body, which is based on the presence of the drug in the system and includes adaptation of the circulating levels of neurotransmitters or hormones. Moreover, as delirium is often associated with DWS, some mechanisms involved in delirium pathogenesis, such as dopamine overflow, acetylcholine deficiency, neuroinflammation and impaired oxidative metabolism [15], have also been considered in the pathophysiology of DWS.

Several diagnostic tools to diagnose DWS have been developed for the pediatric population, such as the Sedation Withdrawal Score (SWS) and the Opioid Benzodiazepine Withdrawal Scale (OBWS), but there are none for adults [9]. Although diagnostic criteria of DWS remain imprecise, signs and symptoms of patients’ discomfort and distress in association with recent reduction or interruption of continuous sedative and analgesic regimens should raise suspicion for DWS [7]. Drug-specific DWS should be differentiated from delirium after weaning off sedation, which is present in almost all patients sedated for more than 3 days and is characterized by temporary altered mentation, fluctuating consciousness and confusion, disorientation, impaired attention and visuospatial ability and cognitive and language disorders [16].

If an abrupt discontinuation or rapid reduction of any sedation-analgesia drug administered for more than 5–7 days is associated with general withdrawal symptoms, such as anxiety, agitation, tremor, tachycardia and hypertension, some agents have specific withdrawal symptoms, i.e., for opioids—mydriasis, nausea/vomiting, abdominal cramps, diarrhea, tachypnea, hot flushes/chills and sweating; for benzodiazepines—fever and seizures; for ketamine—hallucinations, nightmares, depersonalization and general distress [17]; for dexmedetomidine—delirium, hypertension, tachycardia and agitation [18] (Table 1). The onset of DWS-related symptoms is prolonged with long-acting drugs or in the case of alteration of metabolization [19].

## 4. How to Manage Sedation and Analgesia Withdrawal Syndrome in COVID-19 Patients?

As the concern about DWS secondary to infusions in the critically ill is relatively new, no randomized controlled trials for its management are available.

### 4.1. Development of an Analgo-Sedation Plan

In the absence of established preventive strategies of DWS (i.e., daily regimen reduction and early MV weaning) for COVID-19-related severe ARDS patients, local protocols should be developed to adjust sedation and analgesia, and early DWS treatment should be considered for patients requiring high daily doses and prolonged drug administration. In our ICU, we have initially defined sedation targets, i.e., a Nursing Instrument for the Communication of Sedation (NICS) score [20] between −1 and 0 whenever possible (which corresponds to Richmond Agitation-Sedation Scale of −3 to 0); in the case of severe ARDS with high driving pressure despite low tidal volume ventilation, or in the case of high respiratory drive and frequent asynchronies, the target NICS score was −2 (or −4 when using RASS score). The initial analgo-sedation regimen included a combination of propofol (with a maximum of 4 mg/kg·h), sufentanil (0.05 to 0.1 mcg/kg·h with boluses of 0.05 to 0.1 mcg/kg a few minutes before painful procedures) and acetaminophen (1 g q6 h). Although propofol infusion syndrome (PRIS) was quite rare in COVID-19 ARDS patients despite prolonged drug administration [21], creatine kinase and triglyceride concentrations were measured twice a week in all patients.

For patients for whom this regimen did not result in desired sedation levels, a multimodal strategy was applied, with two objectives: opioids sparing, as their efficacy to control the respiratory drive in ARDS patients is low [22] while the risk of intestinal hypomotility is considerable; and limiting benzodiazepine use. Ketamine, which has sedative, analgesic and anti-hyperalgesic properties, is the preferred choice in this setting (at 0.1 to 1 mg/kg·h) [23]. Alpha-2 receptor agonists, i.e., clonidine (at 0.01 to 0.03 mcg/kg·min) or dexmedetomidine (at 0.1 to 1.4 mcg/kg·h) are also effective, as they provide sedation, analgesia and potentiate effects of other drugs (Figure 1) [24]. Only in patients for whom the respiratory drive remained high, or in whom drug-related complications occurred (i.e., liver injury with ketamine; hypotension and/or bradycardia with alpha-2 receptor agonists), benzodiazepines (i.e., midazolam, from 2 to 10 mg/h) were initiated.

### 4.2. Weaning Plan

Once lung mechanics and gas exchanges improved, ventilation by pressure support was initiated and sedation was therefore titrated to limit high tidal volume and high respiratory rate; for patients who had been treated for at least 7 days and were at risk of DWS, daily drug regimens were progressively reduced as follows: propofol 1 mg/kg per 12 h; midazolam 1 mg per 12 h; sufentanil 0.025 mcg/kg per 12 h. If withdrawal symptoms appeared, this was considered to be the “plateau” phase and drug regimens remained unchanged. Alpha-2-receptor agonists were then titrated to higher regimens if tolerated.

### 4.3. Treatment of DWS

Several algorithms have been proposed, mostly based on the treatment of withdrawal syndrome in chronic use or drug abusive disorders. General strategies applied in ICU patients are to shift continuous IV to discontinued oral administration, long half-life molecules to shorter half-life ones and either adjunction of different receptor agonists or auto-receptor feedback inhibitors. Oral equivalents of each drug (i.e., lorazepam for midazolam, methadone for opioids and clonidine for alpha-2-agonists) were introduced; all these medications were then reduced gradually over several weeks.

Alpha-2 adrenergic agonists, i.e., dexmedetomidine or clonidine, bind the α2-subunit, and decrease the release of endogenous catecholamine, thus reducing the adrenergic hyperactivity generated by DWS. Their specific action on the nucleus coeruleus induces sedation and attenuates agitation. They have shown clinical superiority to placebo in reduction of withdrawal symptoms’ severity and treatment completion [25]. As dexmedetomidine has a 6-fold higher affinity for the α2-subunit than clonidine, a hepatic versus renal metabolism, and its elimination half-life is much shorter (2 h versus 6–10 h), it is amenable to frequent titration allowing doses and effects adjustments [26]. Clonidine has similar pharmacologic properties, but its high oral bioavailability, longer half-life, facility of oral method of administration and lower cost provide a convenient and tolerable tapering option for patients on prolonged dexmedetomidine infusions [27]. The pathophysiological mechanism of alpha-2-agonists withdrawal is poorly described [28].

When opioid agonists bind the μ-opioid receptor of noradrenergic neurons, also in the locus coeruleus, the release of norepinephrine is suppressed, resulting in sedation. However, in the event of prolonged exposure, this pathway recovers (i.e., tolerance). After abrupt discontinuation, cAMP synthesis and protein kinase signaling are enhanced, leading to increased norepinephrine release from the locus coeruleus, which underlies some of the characteristic symptoms of opioid withdrawal [29]. Two molecules are available and can facilitate the weaning process while allowing underlying neuroadaptations to revert gradually to their normal state: buprenorphine and methadone, both of which seem comparable in efficacy and duration of withdrawal [30]. Buprenorphine is a high-affinity μ-opioid receptor partial agonist. The elimination half-life is 25–31 h. Its hepatic metabolism limits the risk of accumulation in renal failure. However, replacement of a full agonist by a partial agonist may result in precipitated opioid withdrawal, in particular if given too soon after administration of a full μ-opioid receptor agonist; careful observation for withdrawal symptoms is essential in this setting. An initial dose of 2 mg should be titrated every hour depending on the persistence of a withdrawal syndrome, up to 14–16 mg per day, and then gradually lowered over 2 weeks [31]. Methadone is a μ-opioid receptor full agonist; it has inhibitory effects on NMDA receptors and serotonin reuptake, with an elimination half-life of 12–24 h. As methadone is equipotent to morphine, the dose to be administered corresponds to the quantity of morphine administered over 24 h. In contrast to buprenorphine, there is no risk of precipitating withdrawal syndrome [32]. Prolonged use of opioids also induces N-methyl-d-aspartate (NMDA) receptor activation which is, together with the downregulation of glutamate receptors, implicated in the imbalance between pronociceptive and antinociceptive pathways. This results in attenuated analgesic effects, aggravated pain behaviors, increased tolerance and opioid-induced hyperalgesia (i.e., abnormally increased sensitivity to pain) and/or allodynia (i.e., painful sensation caused by a stimulus that does not normally elicit pain) [33]. This condition is not a DWS, although it may aggravate symptoms if the latter is present. It has to be identified and treated promptly. For the treatment of acute hyperalgesia or allodynia, low-dose boluses of ketamine are an option (0.1 mg/kg) [34]. Pregabalin (i.e., short onset of action, 25 mg q12 h to increase by 25 mg/day, preferably in the evening), by its structure similar to gamma-amino-butyric acid, enhances the action of GABA receptors, and its agonist action on alpha-2-delta receptors decreases the release of glutamate and activates the descending adrenergic pathways.

The biological causes of benzodiazepine withdrawal are multiple and not yet fully understood. Three mechanisms seem important: firstly, prolonged exposure leads to upregulation of GABA-alpha receptors, contributing to the symptoms of tolerance, dependence and withdrawal; secondly, abrupt discontinuation of benzodiazepines is associated with overexpression of NMDA receptors in the hippocampus, increasing brain activity; finally, the activity of the hypothalamic-pituitary adrenocortical axis is amplified, with significant increase of adreno-cortico-trophic hormone (ACTH) and corticosterone plasma levels. Thus, rapid discontinuation of benzodiazepines corresponds to reduced activity of inhibitory GABA functions and a surge in excitatory nervous activity [35]. As for weaning, a shift towards oral lorazepam from IV midazolam was tested with an acceptable security profile in the pediatric ICU population [36]. Comparable to opioids, a gradual reduction over several weeks may be the safest therapeutic option.

Little is known about the mechanism of ketamine withdrawal. A powerful antagonist of NMDA receptor, its onset and clearance are rapid with elimination half-life of 2 h; however, it may be prolonged in the case of liver dysfunction and continuous administration. Signs of withdrawal may be the result of NMDA receptor overexpression following abrupt discontinuation, combined with cognitive disturbances induced by glutamate toxicity in these vulnerable NMDA-overexpressed neurons [37]. Thus, progressive dose reduction appears to be a safe approach. No specific treatment was reported for ketamine withdrawal; cardiovascular agents and unique/small doses of benzodiazepines can be used for treatment of autonomic hyperreactivity and acute agitation [17] (Table 1).

If hyperactive delirium occurs despite application of the recognized bundles of care and non-drug therapies [38], a symptomatic treatment by an oral neuroleptic drug may be initiated, in the absence of contraindication (i.e., prolonged QT interval or extrapyramidal syndrome). Conventional antipsychotic drugs, such as haloperidol, have high affinity for dopamine D2 receptors, markedly interfere with dopaminergic neurotransmission and carry relatively high risk of extrapyramidal symptoms. The atypical antipsychotics have greater affinity for serotoninergic 5-HT2A receptors than for dopamine D2 receptors, and present lower risk of extrapyramidal syndrome [39]. The choice of the molecule should be made with respect to the patient’s condition: for urgent treatment of agitation, only haloperidol allows a rapid effect (i.e., short one-set and half-life; titration by 0.5 mg PO), while other neuroleptics are suitable for intermittent, lower intensity agitation with specific profiles: quietapine (i.e., short half-life; start at 25 to 50 mg q12 h for a maximum of 600–800 mg/day) or olanzapine (i.e., longer half-life, start at 2.5 to 5 mg per day for a maximum of 20 mg/day) if a sedative effect is desired; amisulpride (start at 50 mg/day for a maximum of 800 mg/day) if an anxiolytic effect is required; risperidone (start at 0.5 mg q12 h for a maximum of 4 mg/day) in the case of hallucinations [40].

## 5. Conclusions

The management of DWS related to prolonged sedative and analgesic administration in COVID-19 patients suffering from severe ARDS requires establishment of a clear therapeutic plan from the early phase of therapy. The multimodal pharmacological approach should be integrated with a close monitoring of sedative requirement in these patients and training of the healthcare team to recognize inadequate drug regimens and/or DWS. Future prospective studies are necessary to demonstrate whether such an approach could effectively reduce the occurrence of DWS, help to provide a more adequate adjustment of sedation and analgesia in ARDS COVID-19 patients and eventually reduce the duration of MV and ICU stay.

## Figures and Tables

**Figure 1 jcm-10-04917-f001:**
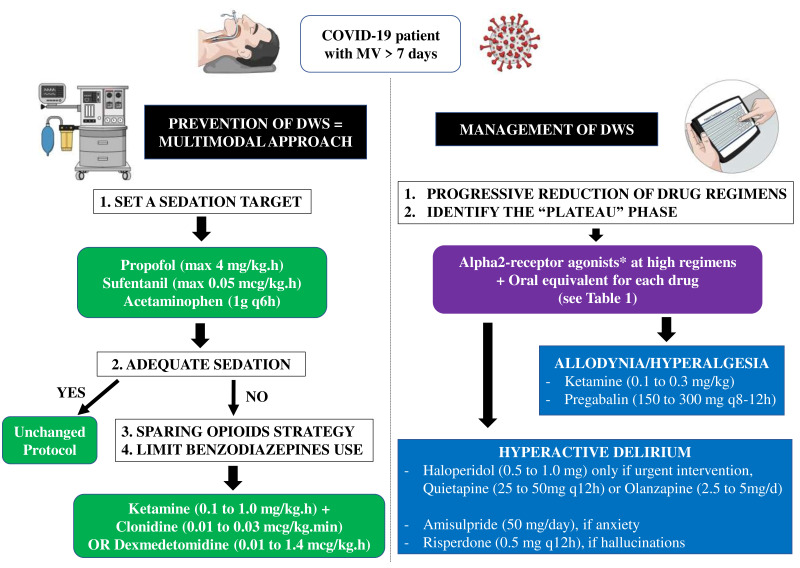
Schematic approach on how to manage sedation and analgesia in COVID-19 patients to minimize the risk or manage drug withdrawal syndrome (DWS). On the left side, a practical approach to prevent DWS is proposed, which includes setting a sedation target using clinical scales (whenever possible) and, in the case of inadequate sedation, avoiding benzodiazepines and minimizing opioids use, with the introduction of ketamine or alpha2-receptor agonists. On the right side, management of DWS is suggested; alpha2-receptor agonists at high regimens (* see text) and shift to oral equivalent for each drug is the first step; in the case of concomitant allodynia/hyperalgesia and/or hyperactive delirium, various drugs can be selected. MV = mechanical ventilation.

**Table 1 jcm-10-04917-t001:** Main general and specific symptoms of drug withdrawal syndrome (DWS) and its specific treatment.

Drug	Symptoms	Specific Treatment
General symptoms	Anxiety, agitation, tremor, tachycardia and hypertension	Neuroleptics Alpha-2-agonists
Opioids	Mydriasis, nausea/vomiting, abdominal cramps, diarrhea, tachypnea, hot flushes/chills, sweating and pain	Buprenorphine Methadone Pregabalin (if hyperalgesia)
Benzodiazepines	Fever and seizures	Lorazepam
Ketamine	Hallucinations, nightmares, depersonalization and general distress	Benzodiazepines Alpha-2-agonists
Dexmedetomidine	Delirium, hypertension, tachycardia and agitation	Clonidine
Propofol	Not described	No specific treatment

## Data Availability

Not applicable.

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
