# Peer review of "How to Manage Withdrawal of Sedation and Analgesia in Mechanically Ventilated COVID-19 Patients?"

_jcm, 2021, doi:10.3390/jcm10214917_

Round 1

Reviewer 1 Report

The authors present a succinct and current review of issues related to withdrawal syndrome (WS) in mechanically ventilated patients treated with analgesics and sedatives.  They acknowledge the sparce data base of interventional studies to treat WS.  However, their review includes an excellent and novel presentation of physiological and pharmacological mechanisms for withdrawal and medications used to treat. I have a few specific questions/comments that might be helpful to the authors:

Lines 58-62:  Please reference the origin(s) of this content related to risk factors.

Lines 72-75:  This sentence is somewhat awkward.  Can you edit for clarity?

Lines 92-96:  Please reference the origin(s) of this content related to onset of symptoms.

Table 1.  Valuable synopsis of information.

Line114:  Please change spelling of Sufentanil.

Figure 1.  Very valuable, but the presentation of content could be improved.  On the left, the two sections in green stand independent of each other.  According to the text, the drugs in the second green section are recommended for consideration if the drugs in the first green section are problematic.  Should you add a statement to that effect in the figure between the two sections?  Regarding to the content on the right of the figure:  (1) you do not include drug doses, like you did on the left.  I recommend consistency; (2) you do not include haloperidol, although it is included in the text; (3) the two blue sections at the bottom are presented as independent.  However, in the text, those on the right (ketamine, pregabalin) are suggested for consideration of the drugs on the left are problematic.  Should you add a statement to that effect in the figure between the two sections? 

Line 138:  Please change spelling of Sufentanil.

Author Response

  1. The authors present a succinct and current review of issues related to withdrawal syndrome (WS) in mechanically ventilated patients treated with analgesics and sedatives.  They acknowledge the sparce data base of interventional studies to treat WS.  However, their review includes an excellent and novel presentation of physiological and pharmacological mechanisms for withdrawal and medications used to treat.

Authors’ response: We thank the reviewer for the nice comment.

  1. Lines 58-62:  Please reference the origin(s) of this content related to risk factors.

Authors’ response: The text has been modified, as requested.

  1. Lines 72-75:  This sentence is somewhat awkward.  Can you edit for clarity?

Authors’ response: The sentence has been edited, accordingly.

  1. Lines 92-96:  Please reference the origin(s) of this content related to onset of symptoms.

Authors’ response: The text has been modified, as requested.

  1. Table 1.  Valuable synopsis of information.

Authors’ response: Many thanks for this.

  1. Line114:  Please change spelling of Sufentanil.

Authors’ response: This has been modified, accordingly.

  1. Figure 1.  Very valuable, but the presentation of content could be improved.  On the left, the two sections in green stand independent of each other.  According to the text, the drugs in the second green section are recommended for consideration if the drugs in the first green section are problematic.  Should you add a statement to that effect in the figure between the two sections?  Regarding to the content on the right of the figure:  (1) you do not include drug doses, like you did on the left.  I recommend consistency; (2) you do not include haloperidol, although it is included in the text; (3) the two blue sections at the bottom are presented as independent.  However, in the text, those on the right (ketamine, pregabalin) are suggested for consideration of the drugs on the left are problematic.  Should you add a statement to that effect in the figure between the two sections? 

Authors’ response: Figure 1 has been modified, as requested.

  1. Line 138:  Please change spelling of Sufentanil.

Authors’ response: This has been modified, accordingly.

Reviewer 2 Report

Concise and well written review. As a narrative review, it presents authors' opinion in an area where high/moderate level evidence is lacking.

No major comments.

Author Response

  1. Concise and well written review. As a narrative review, it presents authors' opinion in an area where high/moderate level evidence is lacking. No major comments.

Authors’ response: We thank the reviewer for the nice comment.